# Understanding the Effects of Data Parallelism and Sparsity on Neural Network Training

**Namhoon Lee**
University of Oxford
namhoon@robots.ox.ac.uk

**Thalaiyasingam Ajanthan**
Australian National University
thalaiyasingama.ajanthan@anu.edu.au

**Philip H. S. Torr**
University of Oxford
phst@robots.ox.ac.uk

**Martin Jaggi**
EPFL
martin.jaggi@epfl.ch

## Abstract

We study two factors in neural network training: data parallelism and sparsity; here, data parallelism means processing training data in parallel using distributed systems (or equivalently increasing batch size), so that training can be accelerated; for sparsity, we refer to pruning parameters in a neural network model, so as to reduce computational and memory cost. Despite their promising benefits, however, understanding of their effects on neural network training remains elusive. In this work, we first measure these effects rigorously by conducting extensive experiments while tuning all metaparameters involved in the optimization. As a result, we find across various workloads of data set, network model, and optimization algorithm that there exists a general scaling trend between batch size and number of training steps to convergence for the effect of data parallelism, and further, difficulty of training under sparsity. Then, we develop a theoretical analysis based on the convergence properties of stochastic gradient methods and smoothness of the optimization landscape, which illustrates the observed phenomena precisely and generally, establishing a better account of the effects of data parallelism and sparsity on neural network training.

## 1 Introduction

Data parallelism is a straightforward and common approach to accelerate neural network training by processing training data in parallel using distributed systems. Being model-agnostic, it is applicable to training any neural networks, and the degree of parallelism equates to the size of mini-batch for synchronized settings, in contrast to other forms of parallelism such as task or model parallelism. While its utility has attracted much attention in recent years, however, distributing and updating large network models at distributed communication rounds still remains a bottleneck (Dean et al., 2012; Hoffer et al., 2017; Goyal et al., 2017; Smith et al., 2018; Shallue et al., 2019; Lin et al., 2020).

Meanwhile, diverse approaches to compress such large network models have been developed, and network pruning – the sparsification process that zeros out many parameters of a network to reduce computations and memory associated with these zero values – has been widely employed (Reed, 1993; Han et al., 2015). In fact, recent studies discovered that pruning can be done at initialization prior to training (Lee et al., 2019; Wang et al., 2020), and by separating the training process from pruning entirely, it not only saves tremendous time and effort in finding trainable sparse networks, but also facilitates the analysis of pruned sparse networks in isolation. Nevertheless, there has been little study concerning the subsequent training of these sparse networks, and various aspects of the optimization of sparse networks remain rather unknown as of yet.

In this work, we focus on studying data parallelism and sparsity[1], and provide clear explanations for their effects on neural network training. Despite a surge of recent interest in their complementary

---

[1]For the purpose of this work, we equate data parallelism and sparsity to increasing batch size and pruning model parameters, respectively; we explain these more in detail in Appendix E.

benefits in modern deep learning, there is a lack of fundamental understanding of their effects. For example, Shallue et al. (2019) provide comprehensive yet empirical evaluations on the effect of data parallelism, while Zhang et al. (2019) use a simple noisy quadratic model to describe the effect; for sparsity, Lee et al. (2020) approach the difficulty of training under sparsity solely from the perspective of initialization.

In this regard, we first accurately measure their effects by performing extensive metaparameter search independently for each and every study case of batch size and sparsity level. As a result, we find a general scaling trend as the effect of data parallelism in training sparse neural networks, across varying sparsity levels and workloads of data set, model and optimization algorithm. Also, the critical batch size turns out to be no less with sparse networks, despite the general difficulty of training sparse networks. We formalize our observation and theoretically prove the effect of data parallelism based on the convergence properties of generalized stochastic gradient methods irrespective of sparsity levels. We take this result further to understand the effect of sparsity based on Lipschitz smoothness analysis, and find that pruning results in a sparse network whose gradient changes relatively too quickly. Notably, this result is developed under standard assumptions used in the optimization literature and generally applied to training using any stochastic gradient method with nonconvex objective and learning rate schedule. Being precise and general, our results could help understand the effects of data parallelism and sparsity on neural network training.

## 2 SETUP

We follow closely the experiment settings used in Shallue et al. (2019). We describe more details including the scale of our experiments in Appendix B, and provide additional results in Appendix D. The code can be found here: https://github.com/namhoonlee/effect-dps-public

**Experiment protocol**.   For a given *workload* (data set, network model, optimization algorithm) and *study* (batch size, sparsity level) setting, we measure the number of training steps required to reach a predefined goal error. We repeat this process for a budget of runs while searching for the best metaparameters involved in the optimization (*e.g.*, learning rate, momentum), so as to record the *lowest* number of steps, namely *steps-to-result*, as our primary quantity of interest. To this end, we regularly evaluate intermediate models on the entire validation set for each training run.

**Workload and study**.   We consider the workloads as the combinations of the followings: (data set) MNIST, Fashion-MNIST, CIFAR-10; (network model) Simple-CNN, ResNet-8; (optimization algorithm) SGD, Momentum, Nesterov with either a fixed or decaying learning rate schedule. For the study setting, we consider a batch size from 2 up to $16384$ and a sparsity level from $0\%$ to $90\%$.

**Metaparameter search**.   We perform a quasi-random search to tune metaparameters efficiently. More precisely, we first generate Sobol low-discrepancy sequences in a unit hypercube and convert them into metaparameters of interest, while taking into account a predefined search space for each metaparameter. The generated values for each metaparameter is in length of the budget of trials, and the search space is designed based on preliminary experimental results.

**Pruning**.   Sparse networks can be obtained by many different ways, and yet, for the purpose of this work, they must not undergo any training beforehand so as to measure the effects of data parallelism while training from scratch. Recent pruning-at-initialization approaches satisfy this requirement, and we adopt the connection sensitivity criterion in Lee et al. (2019) to obtain sparse networks.

## 3 EXPERIMENTAL RESULTS

### 3.1 MEASURING THE EFFECT OF DATA PARALLELISM

First of all, we observe in each and every sparsity level across different workloads a general scaling trend in the relationship between batch size and steps-to-result for the effects of data parallelism (see the 1st and 2nd columns in Figure 1): Initially, we observe a period of *linear scaling* where doubling the batch size reduces the steps to achieve the goal error by half (*i.e.*, it aligns closely with the dashed line), followed by a region of *diminishing returns* where the reduction in the required number of steps by increasing the batch size is less than the inverse proportional amount (*i.e.*, it starts to digress from the linear scaling region), which eventually arrives at a *maximal data parallelism* (*i.e.*, it hits a

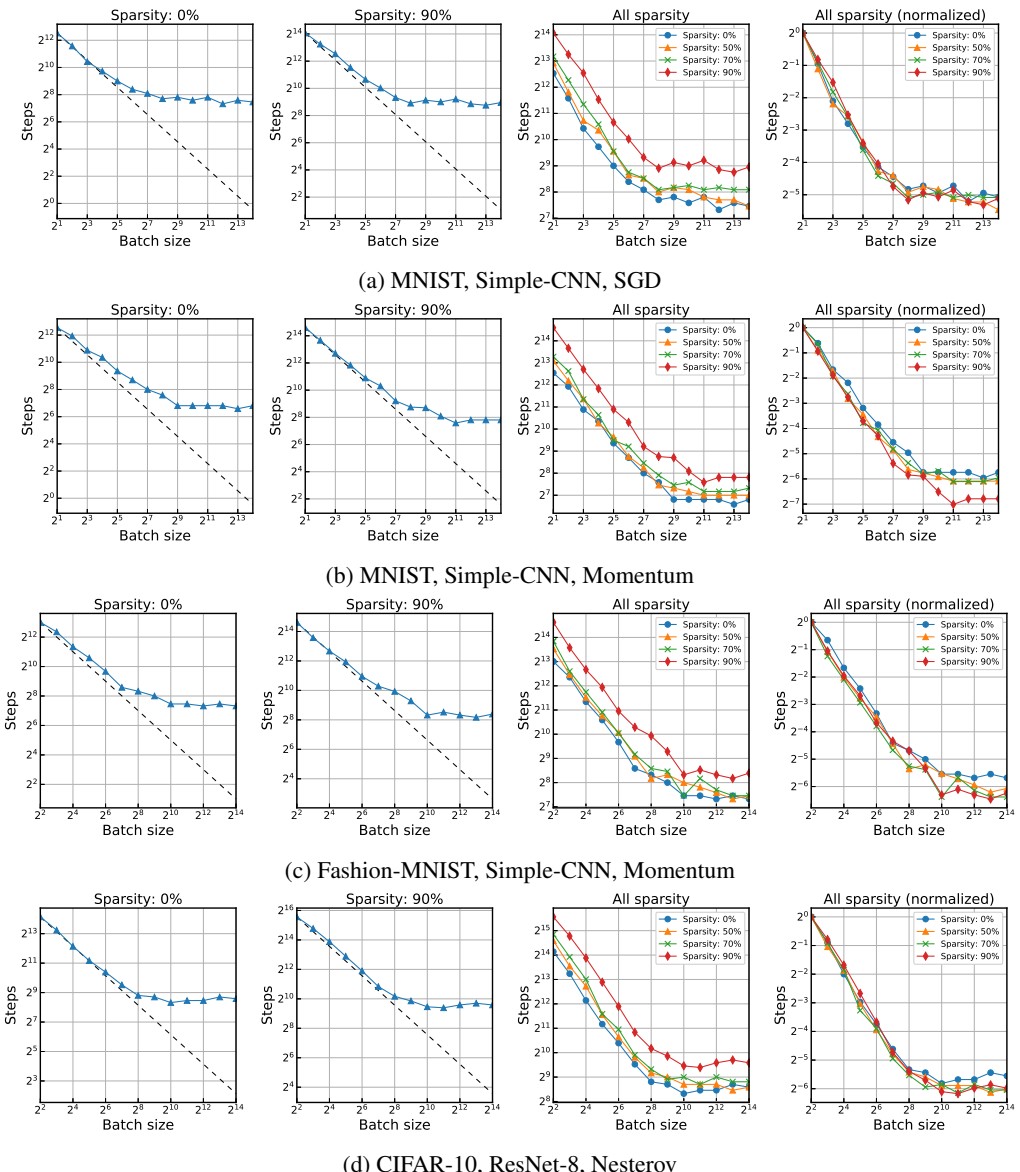

(a) MNIST, Simple-CNN, SGD

(b) MNIST, Simple-CNN, Momentum

(c) Fashion-MNIST, Simple-CNN, Momentum

(d) CIFAR-10, ResNet-8, Nesterov

Figure 1: Effects of data parallelism and sparsity on neural network training for various workloads with a fixed (a-c) or decaying learning rate (d). Across all workloads and sparsity levels, the same scaling pattern is observed for the relationship between batch size and steps-to-result: it starts with the initial phase of *linear scaling*, followed by the region of *diminishing returns*, and eventually reaches to *maximal data parallelism*. Also, the effect of data parallelism in training sparse networks is no worse than that of the dense counterpart, despite the general difficulty of training the former. When training using a *momentum* based SGD, the breakdown of the linear scaling regime often occurs much later at larger batch sizes for a network with higher sparsity. For example, in the case of workload {MNIST, Simple-CNN, Momentum}, the critical batch size for the sparsity 90% network is around $2^{11}$ whereas it is $2^9$ for the sparsity 0% network (see the 4th column in row (b)). This potentially indicates that one can exploit large batch sizes more effectively when training sparse networks than densely parameterized networks. We supply more results in Appendix D.

plateau) where increasing the batch size no longer decreases the steps to reach a goal error. The same trend is observed across various workloads of data set, network model, and optimization algorithm as well as different goal errors (see Appendix D). We note that our observation is consistent with the results of regular network training presented in Shallue et al. (2019); Zhang et al. (2019).

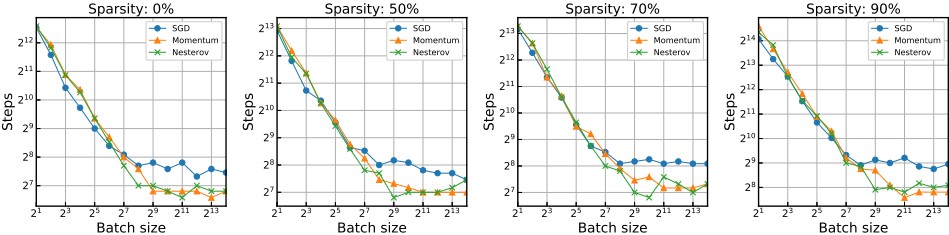

Figure 2: Comparing different optimization algorithms for the effects of data parallelism and sparsity on the workload {MNIST, Simple-CNN, SGD/Momentum/Nesterov} and study {batch size (2-16384), sparsity levels (0, 50, 70, 90%)} settings; there is no normalization or averaging. Across all sparsity levels, momentum optimizers (*i.e.*, Momentum, Nesterov) record lower steps-to-result in a large batch regime and have much bigger critical batch sizes than SGD without momentum. Identifying such patterns is crucial especially when training in resource constrained environments, as practitioners can potentially benefit from reducing the training time by deciding a critical batch size properly, while utilizing resources more effectively.

When we put the results for all sparsity levels together, we observe that training a sparser network takes a longer time; a data parallelism curve for higher sparsity usually lies above than that for lower sparsity (see the 3rd column in Figure 1). For example, compared to the case of sparsity 0% (*i.e.*, the dense, over-parameterized network), 90% sparse network takes about $2 - 4$ times longer training time (or the number of training steps), consistently across different batch sizes (see Figure 12 in Appendix D.1 for more precise comparisons). Recall that we tune all metaparameters independently for each and every study case of batch size and sparsity level without relying on a single predefined training rule in order to find the best steps-to-result. Therefore, this result on the one hand corroborates the general difficulty of training sparse neural networks against the ease of training overly parameterized neural networks.

On the other hand, when we normalize the y-axis of each plot by dividing by the number of steps at the first batch size, we can see the phase transitions more clearly. As a result, we find that the regions of diminishing returns and maximal data parallelism appear no earlier when training sparse networks than the dense network (see the 4th column in Figure 1). This is quite surprising in that one could have easily guessed that the general optimization difficulty incurred by sparsification may influence the data parallelism too, at least to some degree; however, it turns out that the effects of data parallelism on sparse network training remain no worse than the dense case. Moreover, notice that in many cases the breakdown of linear scaling regime occurs even much later at larger batch sizes for a higher sparsity case; this is especially evident for Momentum and Nesterov optimizers (*e.g.*, compare training 90% sparse network using Momentum against 0% dense network). In other words, for sparse networks, a *critical batch size* can be larger, and hence, when it comes to training sparse neural networks, one can increase the batch size (or design a parallel computing system for distributed optimization) more effectively, while better exploiting given resources. We find this result particularly promising since SGD with momentum is often the method of choice in practice.

We further show that momentum optimizers being capable of exploiting large batch sizes hold the same across different sparsity levels by displaying all plots together in Figure 2. Overall, we believe that it is important to confirm the robustness of the data parallelism in sparse neural network training, which has been unknown thus far and difficult to estimate a priori.

## 3.2 ANALYZING METAPARAMETER SEARCH

In this section, we analyze the metaparameter search used to measure the effect of data parallelism. We specifically investigate the workload {MNIST, Simple-CNN, Momentum} where there are two metaparameters to tune (*i.e.*, learning rate and momentum), to visualize all metaparameters easily in a 2D figure (see Appendix D.2 for other results). The results are presented in Figure 3, and we summarize our key findings below:

- Our quasi-random search samples metaparameters efficiently, so that they are distributed evenly (without being cluttered in a log-space) and flexibly (rather than sitting in a grid with fixed spac-

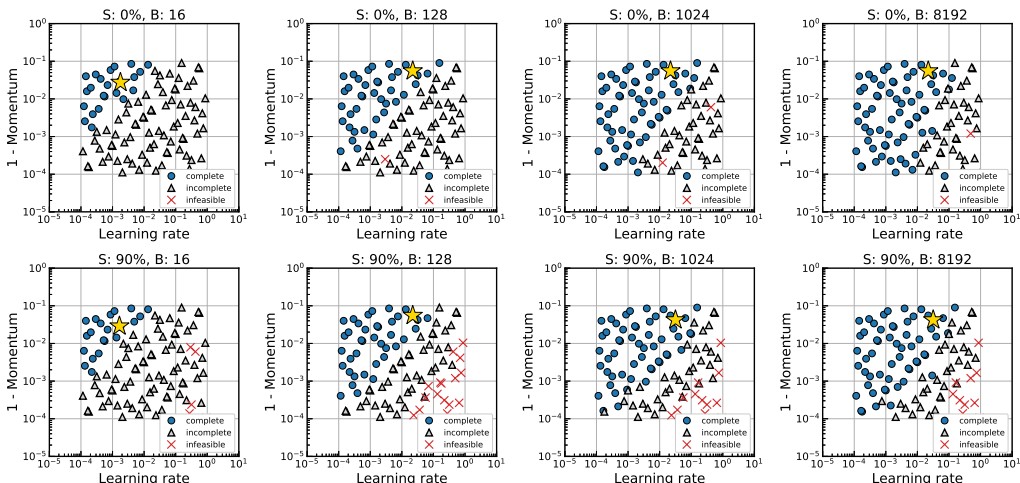

Figure 3: Metaparameter search results (100 samples in total) for Simple-CNN on MNIST trained using Momentum optimizer. Sparsity level (S) and batch size (B) are denoted at the top of each plot. The best trial that records the lowest steps to reach the goal (*i.e.*, steps-to-result) is marked by gold star (⋆). Complete/incomplete refer to the trials of goal reached/not reached given a maximum training step budget, while infeasible refers to the trial of divergence during training.

ing) within the search spaces. Also, the best metaparameters to yield lowest steps (marked by gold star ⋆) are located in the middle of the search ranges rather than sitting at the search boundaries across different batch sizes and sparsity levels. This means that our experiments are designed reasonably well, and the results are reliable.

- There are two distinguished regions (*i.e.*, complete (•) and incomplete (▲)) being separated by a seemingly linear boundary as per the relationship between learning rate and momentum. This indicates that the optimization is being done by an interplay between these two metaparameters; if one metaparameter is not chosen carefully with respect to the other (*e.g.*, increase learning rate for fixed momentum), the optimizer may be stuck in a region spending time oscillating and eventually results in incomplete runs. This highlights the importance of performing metaparameter search, although it is expensive, rather than relying on predetermined heuristic training strategies, in order to accurately measure the effect of data parallelism and avoid potentially suboptimal results.

- The successful region (filled with blue circles •) becomes larger as with increasing batch size, showing that large batch training reaches a given goal error in less number of training iterations than small batch training, and hence, yields more complete runs. Notably, the best learning rate tends to increase as with increasing batch size too. This aligns well with the classic result in learning theory that large batch training allows using bigger learning rates (Robbins & Monro, 1951; Bottou, 1998; Krizhevsky, 2014).

## 4 UNDERSTANDING THE EFFECTS OF DATA PARALLELISM AND SPARSITY

So far, we have focused on measuring the effects of data parallelism and sparsity on neural network training, and as a result found two distinctive global phenomena across various workloads: scaling trend between batch size and steps-to-result, and training difficulty under sparsity. While our findings align well with previous observations (Shallue et al., 2019; Zhang et al., 2019; Lee et al., 2020), it remains unclear as to why it occurs, and whether it will generalize. To this end, we establish theoretical results that precisely account for such phenomena based on convergence properties of generalized stochastic gradient methods in this section.

### 4.1 CONVERGENCE ANALYSIS FOR THE GENERAL EFFECTS OF DATA PARALLELISM

Let us begin with reviewing the convergence properties of stochastic gradient methods as the choice of numerical algorithms for solving optimization problems. Consider a generic optimization problem where the objective is to minimize empirical risk with the objective function $f : \mathbb{R}^m \to \mathbb{R}$, a

prediction function $h : \mathbb{R}^{d_x} \times \mathbb{R}^m \to \mathbb{R}^{d_y}$, and a loss function $l : \mathbb{R}^{d_y} \times \mathbb{R}^{d_y} \to \mathbb{R}$ which yields the loss $l(h(\boldsymbol{x}; \mathbf{w}), \boldsymbol{y})$ given an input-output pair $(\boldsymbol{x}, \boldsymbol{y})$, where $\mathbf{w} \in \mathbb{R}^m$ is the parameters of the prediction model $h$, and $d_x$ and $d_y$ denote the dimensions of input $\boldsymbol{x}$ and output $\boldsymbol{y}$, respectively. A generalized stochastic gradient method to solve this problem can be of the following form:

$$\mathbf{w}_{k+1} := \mathbf{w}_k - \eta_k g(\mathbf{w}_k, \boldsymbol{\xi}_k), \tag{1}$$

where $\eta_k$ is a scalar learning rate, $g(\mathbf{w}_k, \boldsymbol{\xi}_k) \in \mathbb{R}^m$ is a stochastic vector (*e.g.*, unbiased estimate of the gradient $\nabla f$) with $\boldsymbol{\xi}_k$ denoting a random variable to realize data samples, either a single sample as in the prototypical stochastic gradient method (Robbins & Monro, 1951) or a set of samples as in the mini-batch version (Bottou, 1991). Given an initial iterate $\mathbf{w}_1$, it finds a solution by performing the above update iteratively until convergence.

Under the assumptions[2] of Lipschitz smoothness of $f$ and bounded variance of $g$, the convergence rate result states that for such generic problem with nonconvex objective and optimization method with a fixed [3] learning rate $\eta_k = \bar{\eta}$ for all $k$ satisfying $0 < \bar{\eta} \le \frac{\mu}{LM_G}$, the expected average squared norm of gradients of the objective function is guaranteed to satisfy the following inequality for all $K \in \mathbb{N}$ (Bottou et al., 2018):

$$\mathbb{E}\left[\frac{1}{K} \sum_{k=1}^{K} \|\nabla f(\mathbf{w}_k)\|_2^2\right] \le \frac{\bar{\eta}LM}{\mu} + \frac{2(f(\mathbf{w}_1) - f_\infty)}{K\mu\bar{\eta}}. \tag{2}$$

Here, $f(\mathbf{w}_1)$, $f_\infty$, $\nabla f(\mathbf{w}_k)$ refer to the objective function's value at $\mathbf{w}_1$, lower bound, gradient at $\mathbf{w}_k$, respectively. Also, $L$ is the Lipschitz constant of $\nabla f$, and $\mu$, $M$, $M_G$ denote scalar bounds in the assumption on the second moment of $g(\mathbf{w}_k, \boldsymbol{\xi}_k)$. Note here that $M$ is linked to batch size $B$ as $M \propto 1/B$. In addition, if $g(\mathbf{w}_k, \boldsymbol{\xi}_k)$ is an unbiased estimate of $\nabla f(\mathbf{w}_k)$, which is the case for $\boldsymbol{\xi}_k$ being i.i.d. samples as in our experiments, then simply $\mu = 1$ (Bottou et al., 2018). In essence, this result shows that the average squared gradient norm on the left-hand side is bounded above by asymptotically decreasing quantity as per $K$, indicating a sublinear convergence rate of the method. We note further that the convergence rate for the mini-batch stochastic optimization of nonconvex loss functions is studied previously (Ghadimi et al., 2016; Wang & Srebro, 2017), and yet, here we reconsider it to analyze the effects of data parallelism.

We now reformulate this result, such that it is translated into a form that matches our experiment settings and reveals the relationship between batch size and steps-to-result. We start by recognizing that the quantity on the left-hand side, the expected average squared norm of $\nabla f(\mathbf{w}_k)$ during the first $K$ iterations, indicates the degree of convergence; for example, it gets smaller as training proceeds with increasing $K$. Thus, this quantity is directly related to a goal error to reach in our experiments, which is set to be fixed across different batch sizes for a given workload. This effectively means that training has stopped, and $K$ will no longer contribute to decrease the bound of the quantity. Also, recall that we select the *optimal* learning rate $\bar{\eta}^\star$, out of extensive metaparameter search, to record the *lowest* number of steps to reach the given goal error, *i.e.*, steps-to-result $K^\star$. Next, notice that the only factors that constitute the inequality in Eq. (2) are the Lipschitz constant $L$ and the variance bound $M$, and if they are assumed to be tight in the worst case, the inequality becomes tight. Now we are ready to provide the relationship between batch size ($B$) and steps-to-result ($K^\star$) as follows:

**Proposition 4.1.** Let $\varepsilon = \mathbb{E}\left[\frac{1}{K^\star} \sum_{k=1}^{K^\star} \|\nabla f(\mathbf{w}_k)\|_2^2\right]$ denote a degree of convergence achieved after the first $K^\star$ iterations and $\bar{\eta}^\star$ denote the optimal learning rate used to yield the lowest number of steps $K^\star$ to reach $\varepsilon$. Then,

$$K^\star \approx \frac{c_1}{B} + c_2, \qquad \text{where} \quad c_1 = \frac{\Delta L \beta}{\mu^2 \varepsilon^2} \quad \text{and} \quad c_2 = \frac{\Delta}{\bar{\eta}^\star \mu \varepsilon}, \tag{3}$$

where $\Delta = 2(f(\mathbf{w}_1) - f_\infty)$, $\beta$ is the initial variance bound at batch size $B = 1$ for a given workload.

*Proof.* This result is obtained by recasting Eq. (2) as outlined above. The proof is in Appendix A. $\square$

---

[2](i) $f$ is differentiable and satisfies $\|\nabla f(\mathbf{w}) - \nabla f(\bar{\mathbf{w}})\|_2 \le L\|\mathbf{w} - \bar{\mathbf{w}}\|_2$, $\forall\{\mathbf{w}, \bar{\mathbf{w}}\} \subset \mathbb{R}^m$, and (ii) there exist scalars $M \ge 0$, $M_G \ge \mu^2 > 0$ such that $\mathbb{E}_{\boldsymbol{\xi}_k}\left[\|g(\mathbf{w}_k, \boldsymbol{\xi}_k)\|_2^2\right] \le M + M_G\|\nabla f(\mathbf{w}_k)\|_2^2$.

[3]We also consider the general decaying learning rate case and prove the same result in Appendix A.2.

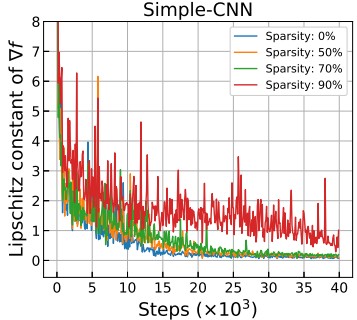 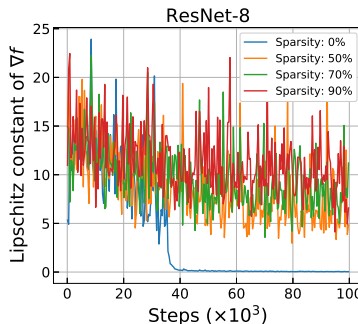

Figure 4: Lipschitz constant of $\nabla f$ measured locally over the course of training for networks with different sparsity levels. The more a network is pruned, the higher the Lipschitz constant becomes; *e.g.*, for 0, 50, 70, 90% sparsity levels, the average Lipschitz constants are 0.57, 0.72, 0.81, 1.76 for Simple-CNN and 3.87, 8.74, 9.54, 11.18 for ResNet-8, respectively. This indicates that pruning results in a network whose gradients are less smooth during training. We further provide additional training logs and explain how smoothness is measured in Appendix C.

This result precisely illustrates the relationship between batch size and steps-to-result. For example, when $B$ is small, $K^\star \approx \frac{c_1}{B}$, fitting the linear scaling regime (*e.g.*, $B \to 2B$ makes $K^\star \to (1/2)K^\star$), whereas when $B$ is large and asymptotically dominates the right-hand side, $K^\star \approx c_2$, indicating the maximal data parallelism as $K^\star$ remains constant. In general, for moderate batch sizes, scaling $B \to 2^r B$ results in $K^\star \to \frac{1}{2^r}K^\star + (1 - \frac{1}{2^r})c_2$ (rather than $\frac{1}{2^r}K^\star$), indicating diminishing returns.

Moreover, we prove the same relationship between $B$ and $K^\star$ (with different constant terms) for the general decaying learning rate case (see Appendix A.2). Therefore, this result not only well accounts for the scaling trend observed in the experiments, but also describes it more precisely and generally. Notably, the effect of data parallelism, which has only been addressed empirically and thus remained as debatable, is now theoretically verified and applicable to general nonconvex objectives. We will further relate our result to sparse networks via smoothness analysis in Section 4.2.

## 4.2  LIPSCHITZ SMOOTHNESS FOR THE DIFFICULTY OF TRAINING SPARSE NETWORKS

Another distinct phenomenon observed in our experiments is that the number of steps required to reach the same goal error for sparse networks is consistently higher than that for dense networks regardless of batch size (*i.e.*, a whole data parallelism curve shifts upwards when introducing sparsity). This indicates the general difficulty of training sparse neural networks, and that sparsity degrades the training speed. In this section, we investigate what may cause this difficulty, and find a potential source of the problem by inspecting our theory of the effect of data parallelism to this end.

Let us begin with our result for the effect of data parallelism in Proposition 4.1. Notice that it is the coefficient $c_1$ ($= \Delta L \beta / \mu^2 \varepsilon^2$) that can shift a whole data parallelism curve vertically, by the same factor across different batch sizes. Taking a closer look, we realize that it is the Lipschitz constant $L$ that can vary quite significantly by introducing sparsity and hence affect $c_1$; $\varepsilon$ and $\mu$ are fixed, and $\Delta$ and $\beta$ can change by sparsity in a relatively minor manner (we explain this in detail in Appendix C). Specifically, $L$ refers to the bound on the rate of change in $\nabla f$ and is by definition a function of $f$. Also, sparsity introduced by pruning changes the prediction function $h$ which is linked to $f$ via the loss function $l$. Therefore, we posit that a sparse neural network obtained by pruning will be *less smooth* (with a higher Lipschitz constant) than the non-pruned dense network. To verify our hypothesis, we empirically measure the Lipschitz constant for networks with different sparsity levels over the course of the entire training process. The results are presented in Figure 4.

As we can see, it turns out that the Lipschitz constant increases as with increasing sparsity level, and further, is consistently higher for sparse networks than for the dense network throughout training. This means that *pruning results in sparse networks whose gradient changes relatively too quickly* compared to the dense network; in other words, the prediction function $h$ becomes *less smooth* after pruning. This is potentially what hinders training progress, and as a result, sparse networks require more time (*i.e.*, steps-to-result) to reach the same goal error.

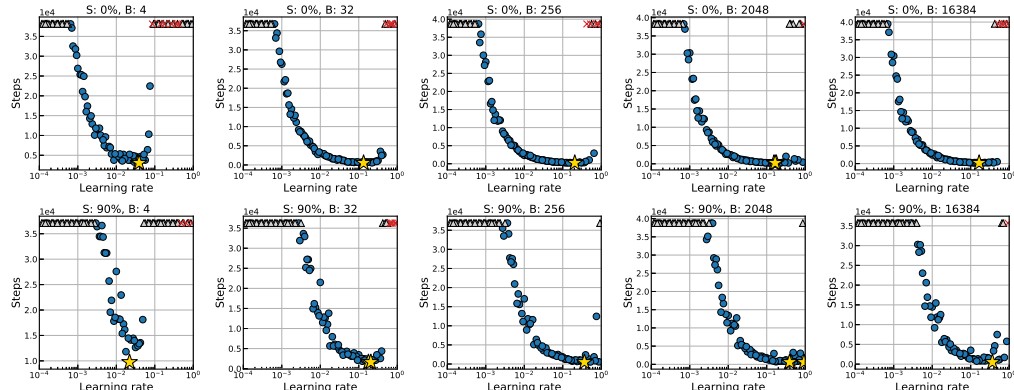

Figure 5: Metaparameter search results for the workload of {MNIST, Simple-CNN, SGD}, where the metaparameter being tuned is the learning rate $\bar{\eta}$. The blue circles (•) denote successful runs (*i.e.*, it reached the goal error), and the best trial that records the steps-to-result is marked by gold star (★); also, the grey triangles (▲) and red crosses (×) refer to incomplete and infeasible runs, respectively. The range of $\bar{\eta}$ shrinks as the sparsity level increases from 0% to 90%, indicating *increased L* based on the learning rate condition from the convergence properties satisfying $0 < \bar{\eta} \leq 1/L$.

Evidence of increased Lipschitz constant for sparse networks can be found further in metaparameter search results presented in Figure 5. Notice that for each batch size, the size of the range for successful learning rate $\bar{\eta}$ decreases when switching from 0 to 90% sparsity level. This is because the learning rate bound satisfying the convergence rate theory becomes $0 < \bar{\eta} \leq 1/L$ for a fixed batch size, and increased $L$ due to sparsity shrinks the range of $\bar{\eta}$.

We note that our findings of increased Lipschitz constant for sparse networks are consistent with the literature on over-parameterized networks such as Li & Liang (2018), which can be seen as the opposite of sparsity. The more input weights a neuron has, the less likely it is that a single parameter significantly changes the resulting activation pattern, and that wide layers exhibit convexity-like properties in the optimization landscape Du et al. (2019). This even extends to non-smooth networks with ReLU activations, which are still shown to exhibit pseudo-smoothness in the overparameterized regime Li & Liang (2018). We further show that our theory precisely explains the difficulty of training sparse networks due to decreased smoothness based on a quantitative analysis in Appendix C.

In addition, we provide in Figure 6 the training logs of the networks used for the Lipschitz smoothness analysis, in order to show the correlation between the Lipschitz smoothness of a network and its training performance; *i.e.*, sparsity incurs low smoothness of gradients (high $L$; see Figure 4) and hence the poor training performance.

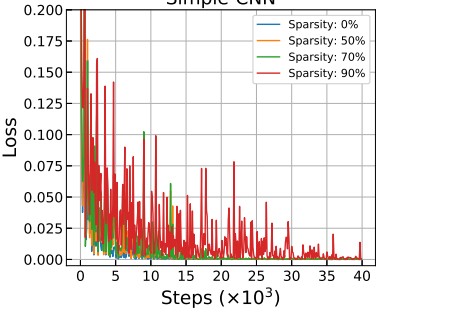
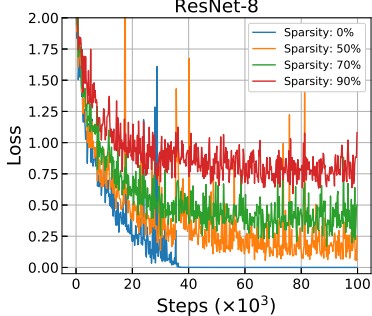

Figure 6: Training logs of Simple-CNN and ResNet-8 used for the smoothness analysis. The sparse networks that recorded high Lipschitz constants show worse training performance, indicating that low smoothness may be the potential cause of hampering the training of sparse neural networks.

## 5 DISCUSSION

Data parallelism with sparsity could have promising complementary benefits, and yet, little has been studied about their effects on neural network training thus far. In this work, we accurately measured their effects, and established theoretical results that precisely account for the general characteristics of data parallelism and sparsity based on the convergence properties of stochastic gradient methods and Lipschitz smoothness analysis. We believe our results are significant, in that these phenomena, which have only been addressed partially and empirically, are now theoretically verified with more accurate descriptions and applied to general nonconvex settings.

While our findings render positive impacts to practitioners and theorists alike, there are remaining challenges. First, our experiments are bounded by available computing resources, and the cost of experiments increases critically for more complex workloads. Also, the lack of convergence guarantees for existing momentum schemes in nonconvex and stochastic settings hinders a further theoretical analysis. We hypothesize that ultimate understanding of the effect of data parallelism should be accompanied by a study of the generalization capability of optimization methods. Nonetheless, these are beyond the scope of this work, and we intend to explore these directions as future work.

### ACKNOWLEDGMENTS

This work was supported by the ERC grant ERC-2012-AdG 321162-HELIOS, EPSRC grant Seebibyte EP/M013774/1, EPSRC/MURI grant EP/N019474/1, the Australian Research Council Centre of Excellence for Robotic Vision (project number CE140100016), and the Institute of Information & communications Technology Planning & Evaluation (IITP) grant funded by the Korea government (MSIT) (No.2020-0-01336, Artificial Intelligence Graduate School Program (UNIST)). We would also like to acknowledge the Royal Academy of Engineering and FiveAI.

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

## A   PROOF OF THE GENERAL EFFECT OF DATA PARALLELISM

This section provides the missing proofs in Section 4. The goal is to derive the relationship between batch size $B$ and steps-to-result $K^\star$ from the convergence rates of generalized stochastic gradient methods for both fixed and decaying learning rate cases. The result serves to account for the effect of data parallelism as a general phenomenon that must appear naturally at neural network training.

### A.1   FIXED LEARNING RATE CASE

We start from the convergence rate result in Eq. (2). We first recognize that the expected average squared gradient norm on the left-hand side indicates the degree of convergence. Then, this quantity is directly related to the concept of goal error to reach in our experiments, and hence, reduces to be a small constant $\epsilon$ as soon as it is implemented as a pre-defined goal error for a given workload. Thus, it follows that

$$\epsilon \leq \frac{\bar{\eta} L M}{\mu} + \frac{2(f(\mathbf{w}_1) - f_\infty)}{K \mu \bar{\eta}} . \tag{4}$$

Notice that by fixing $\epsilon = \mathbb{E}\left[\frac{1}{K} \sum_{k=1}^{K} \|\nabla f(\mathbf{w}_k)\|_2^2\right]$, it effectively means that the training process has stopped, and therefore, $K$ on the right-hand side will no longer contribute to decrease the bound of the quantity for a particular learning rate $\bar{\eta}$ and batch size $B$.

Also, we select the *optimal*[4] learning rate $\bar{\eta}^\star$, out of extensive metaparameter search, to record the *lowest* number of steps to reach the given goal error, which we denote as steps-to-result $K^\star$.

---

[4]Here, *optimal* simply refers to the sense of yielding the *lowest* steps-to-result.

Plugging in these yields the following:

$$\epsilon \leq \frac{\bar{\eta}^{\star} L M}{\mu} + \frac{2(f(\mathbf{w}_1) - f_{\infty})}{K^{\star} \mu \bar{\eta}} .$$ (5)

Next, notice that the only factors that constitute the inequality in Eq. (2) come from the assumptions made to derive the convergence rate result, which are on the Lipschitz smoothness $L$ and the variance bound $M$, and if they are assumed to be tight in the worst case, the inequality becomes tight. Then, after making algebraic manipulation and taking the first-order Taylor approximation while substituting $M = \beta/B$ since the variance bound $M$ is related to batch size $B$ as $M \propto 1/B$ (Bottou et al., 2018), we obtain the following result:

$$K^{\star} \approx \frac{\Delta}{\bar{\eta}^{\star} \mu \epsilon - (\bar{\eta}^{\star})^2 L M}$$ (6)

$$\approx \frac{\Delta}{\bar{\eta}^{\star} \mu \epsilon} + \frac{\Delta L M}{\mu^2 \epsilon^2}$$

$$= \frac{\Delta L \beta}{\mu^2 \epsilon^2 B} + \frac{\Delta}{\bar{\eta}^{\star} \mu \epsilon}$$

$$= \frac{c_1}{B} + c_2 , \qquad \text{where } c_1 = \frac{\Delta L \beta}{\mu^2 \epsilon^2} \text{ and } c_2 = \frac{\Delta}{\bar{\eta}^{\star} \mu \epsilon} .$$

Here, $\Delta = 2(f(\mathbf{w}_1) - f_{\infty})$, $\beta$ is the initial variance bound at batch size $B = 1$ for a given workload. Notice that $\epsilon$, $\Delta$, $L$, $\beta$, $\mu$ $\bar{\eta}^{\star}$ all are constant or become fixed for a given workload. Also, the degree of metaparameter search quality is assumed to be the same across different batch sizes, and hence, the result can be reliably used to interpret the relationshp between batch size and steps-to-result.

## A.2 DECAYING LEARNING RATE CASE

We could extend the convergence rate for a fixed learning rate $\eta_k = \bar{\eta}$ in Eq. (2) to any sequence of decaying learning rates $\eta_k$ satisfying $\sum_{k=1}^{\infty} \eta_k = \infty$ and $\sum_{k=1}^{\infty} \eta_k^2 < \infty$ based on Theorem 4.10 in Bottou et al. (2018) as follows:

$$\mathbb{E}\left[\frac{1}{K} \sum_{k=1}^{K} \eta_k \|\nabla f(\mathbf{w}_k)\|_2^2\right] \leq \frac{L M}{K \mu} \sum_{k=1}^{K} \eta_k^2 + \frac{2(\mathbb{E}[f(\mathbf{w}_1)] - f_{\infty})}{K \mu} .$$ (7)

The requirements on learning rate $\eta_k$ are the classical conditions (Robbins & Monro, 1951) that are assumed for convergence of any (sub)gradient methods, and cover nearly all standard and/or heuristical learning rate schedules employed in practice.

Now, the relationship between batch size and steps-to-result can be derived similarly as before. First, applying $\tilde{\epsilon} = \mathbb{E}\left[\frac{1}{K} \sum_{k=1}^{K} \eta_k \|\nabla f(\mathbf{w}_k)\|_2^2\right]$ for the degree of convergence, and replacing with $\Delta = 2(\mathbb{E}[f(\mathbf{w}_1)] - f_{\infty})$ for simplicity, Eq. (7) can be rewritten as follows:

$$\tilde{\epsilon} \leq \frac{L M}{K \mu} \sum_{k=1}^{K} \eta_k^2 + \frac{\Delta}{K \mu} .$$ (8)

Plugging $H = \sum_{k=1}^{K} \eta_k^2$ for a finite constant from decaying learning rate, and further, $H^{\star}$ and $K^{\star}$ for selected $H$ by metaparameter search and steps-to-result, respectively, it becomes:

$$\tilde{\epsilon} \leq \frac{L M H^{\star}}{K^{\star} \mu} + \frac{\Delta}{K^{\star} \mu} .$$ (9)

Finally, using the worst case tightness on $L$ and $M$, substituting $M = \beta/B$, and rearranging the terms, the effect of data parallelism for decaying learning rate case can be written as follows:

$$K^{\star} \approx \frac{L M H^{\star}}{\mu \tilde{\epsilon}} + \frac{\Delta}{\mu \tilde{\epsilon}}$$ (10)

$$= \frac{L H^{\star} \beta}{\mu \tilde{\epsilon} B} + \frac{\Delta}{\mu \tilde{\epsilon}}$$

$$= \frac{\tilde{c}_1}{B} + \tilde{c}_2 , \qquad \text{where } \tilde{c}_1 = \frac{L H^{\star} \beta}{\mu \tilde{\epsilon}} \text{ and } \tilde{c}_2 = \frac{\Delta}{\mu \tilde{\epsilon}} .$$

Here, $\tilde{\epsilon}$, $\Delta$, $L$, $H^{\star}$, $\beta$, $\mu$ all are constant or become fixed for a given workload.

This result, along with the case of fixed learning rate, establishes the theoretical account for the effect of data parallelism, by precisely and generally describing the relationship between batch size $B$ and steps-to-result $K^{\star}$. Further analysis of these results is referred to the main paper.

## B  SCALE OF OUR EXPERIMENTS

For a given workload of {data set, network model, optimization algorithm} and for a study setting of {batch size, sparsity level}, we execute 100 training runs with different metaparameters to measure steps-to-result. At each run, we evaluate the intermediate models at every 16 (for MNIST) or 32 (for CIFAR-10) iterations, on the *entire* validation set, to check if it reached a goal error. This means that, in order to plot the results for the workload of {MNIST, Simple-CNN, SGD} for example, one would need to perform, 14 (batch sizes; $2^1$ to $2^{14}$) $\times$ 4 (sparsity levels; $0, 50, 70, 90\%$) $\times$ 100 (runs) $\times$ $40,000$ (max training iteration preset) / 16 (evaluation interval) $= 14,000,000$ number of evaluations. Assuming that evaluating the Simple-CNN model on the entire MNIST validation set takes only a *second* on a modern GPU, it will take $14,000,000$ (evaluations) $\times$ 1 (second per evaluation) / 3600 (second per hour) $\approx 3888$ hours or 162 days.

Of course, there are multiple ways to reduce this cost; for instance, we may decide to stop as soon as the run hits the goal error without running until the max training iteration limit. Or, simply reducing any factor listed above that contributes to increasing the experiment cost (*e.g.*, number of batch sizes) can help to reduce the time, however, in exchange for the quality of experiments. We should also point out that this is only for one workload where we assumed that the evaluation takes only a second. The cost can increase quite drastically if the workload becomes more complex and requires more time for evaluation and training (*e.g.*, CIFAR-10). Not to mention, we have tested for multiple workloads besides the above example, in order to confirm the generality of the findings in this work.

## C  MORE ON LIPSCHITZ SMOOTHNESS ANALYSIS

We measure the local Lipschitz constant of $\nabla f$ based on a Hessian-free method as used in Zhang et al. (2020). Precisely, the local smoothness (or Lipschitz constant of the gradient) at iteration $k$ is estimated as in the following:

$$\hat{L}(\mathbf{w}_k) = \max_{\gamma \in \{\delta, 2\delta, .., 1\}} \frac{\|\nabla f(\mathbf{w}_k + \gamma \boldsymbol{d}) - \nabla f(\mathbf{w}_k)\|_2}{\|\gamma \boldsymbol{d}\|_2} \,, \tag{11}$$

where $\boldsymbol{d} = \mathbf{w}_{k+1} - \mathbf{w}_k$ and $\delta \in (0, 1)$ for which we set to be $\delta = 0.1$. The expected gradient $\nabla f$ is computed on the entire training set, and we measure $\hat{L}(\mathbf{w}_k)$ at every 100 iterations throughout training. This method searches the maximum bound on the smoothness along the direction between $\nabla f(\mathbf{w}_{k+1})$ and $\nabla f(\mathbf{w}_k)$ based on the intuition that the degree of deviation of the linearly approximated objective function is bounded by the variation of gradient between $\mathbf{w}_{k+1}$ and $\mathbf{w}_k$. Furthermore, while ReLU networks (*e.g.*, Simple-CNN, ResNet-8) can only be piecewise smooth, the smoothness can still be measured for the same reason that we measure gradient (*i.e.*, it only requires differentiability).

We also empirically measure the changes in $\Delta$ and $\beta$ by introducing sparsity. Recall that these are the other elements in $c_1$ that can be affected by sparsity along with Lipschitz constant $L$. When we measure these quantities for Simple-CNN, we obtain the following results: $\Delta_s / \Delta_d \approx 4.68/4.66 \approx 1.00$, and $\beta_s / \beta_d \approx 107.39/197.06 \approx 0.54$; more precisely, $\Delta$ does not change much since neither $f(\mathbf{w}_1)$ or $f(\mathbf{w}_\infty)$ changes much, and $\beta_s / \beta_d$ can be measured by the ratio of the variances of gradients between sparse and dense networks at $B = 1$. We have already provided $L_s$, $L_d$ in Figure 4, which makes $L_s / L_d \approx 1.76/0.57 \approx 3.09$. Here, s and d denote sparse (90%) and dense, respectively. Notice that if we combine all the changes in $\Delta$, $\beta$, $L$ due to sparsity, and compute $c_{1,s}/c_{1,d}$, it becomes $1.00 \times 0.54 \times 3.09 \approx 1.67$. Importantly, $c_{1,s}/c_{1,d} > 1$ means that $c_1$ has increased by sparsity, and since the increase in Lipschitz constant $L$ played a major role therein, these results indicate that the general difficulty of training sparse networks is indeed caused by reduced smoothness. We further note that this degree of change fits roughly the range of $k_s^\star/k_d^\star$ as shown in Figure 12.

# D  ADDITIONAL RESULTS

In this section, we provide additional experiemical results that are not included in the main paper. In Section D.1, we supplement more results for the effects of data parallelism and sparsity in Figures 7, 8, 9, 10, 11. In Figure 12 we present the difference in ratio between sparse (90%) and dense networks across different batch sizes for all workloads presented in this work. This result shows how much increase in steps-to-result is induced by introducing sparsity, and therefore, is used to study the general difficulty of training sparse neural networks. In Section D.2, we provide metaparameter search results for a subset of workloads studied in this work.

## D.1  EFFECTS OF DATA PARALLELISM AND SPARSITY

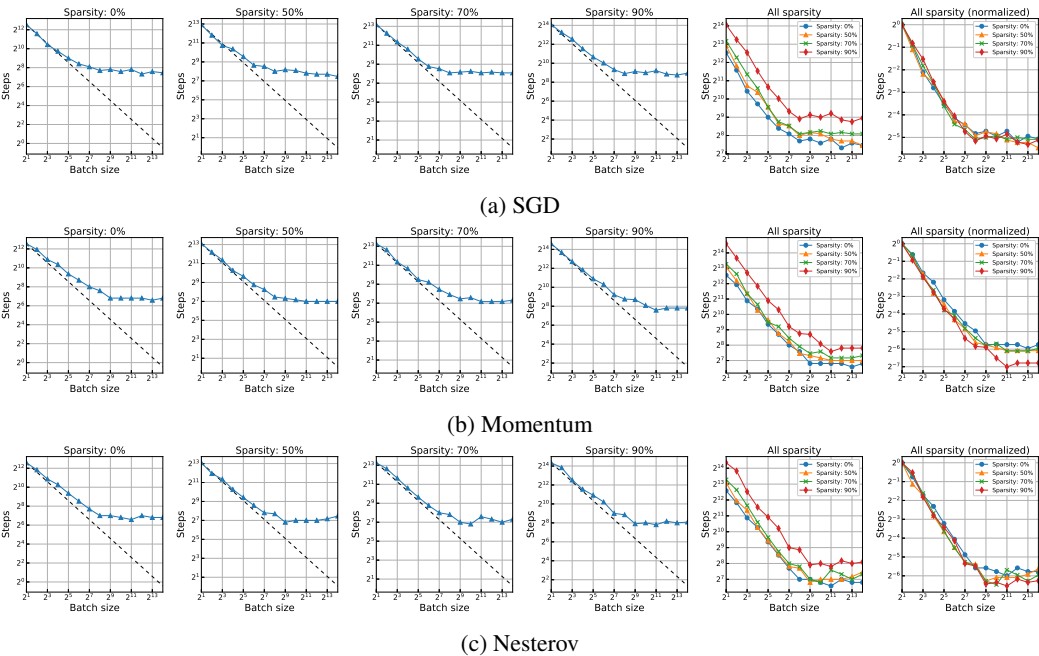

(a) SGD

(b) Momentum

(c) Nesterov

Figure 7: Results for the effects of data parallelism for the workloads of {MNIST, Simple-CNN, SGD/Momentum/Nesterov} with a constant learning rate.

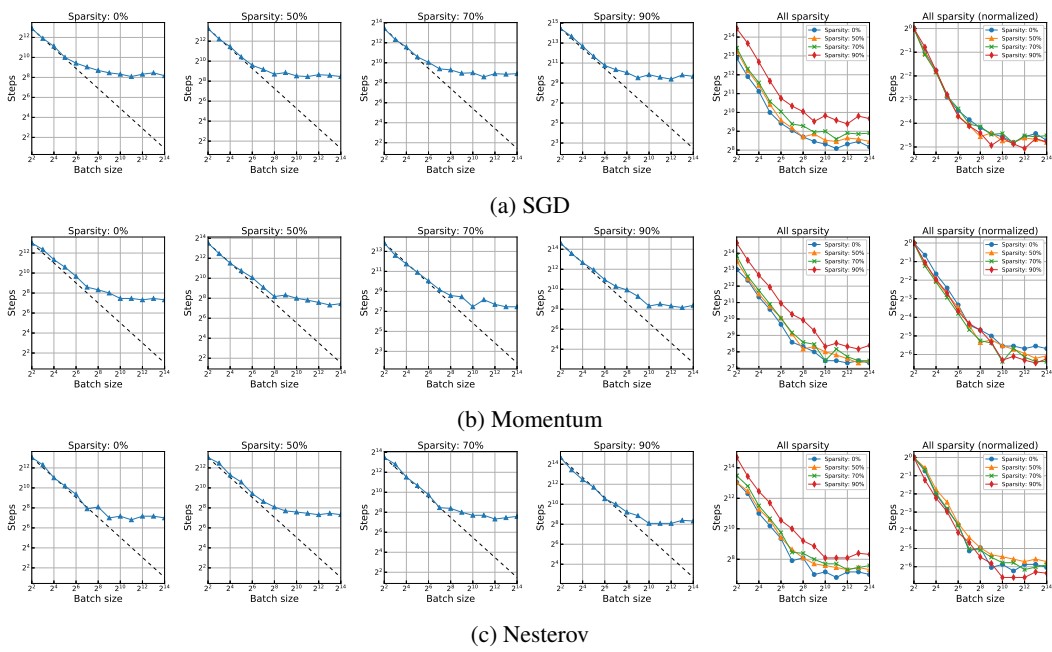

Figure 8: Results for the effects of data parallelism for the workloads of {Fashion-MNIST, Simple-CNN, SGD/Momentum/Nesterov} with a constant learning rate and the goal error of $0.12$.

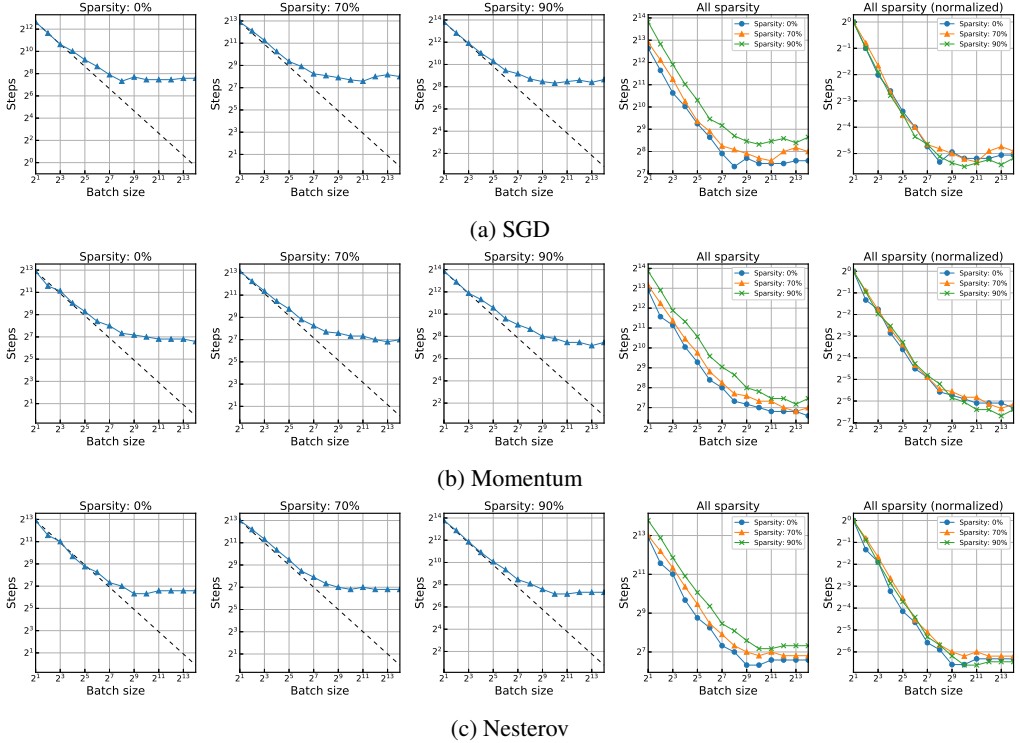

Figure 9: Results for the effects of data parallelism for the workloads of {Fashion-MNIST, Simple-CNN, SGD/Momentum/Nesterov} with a constant learning rate and the goal error of $0.14$.

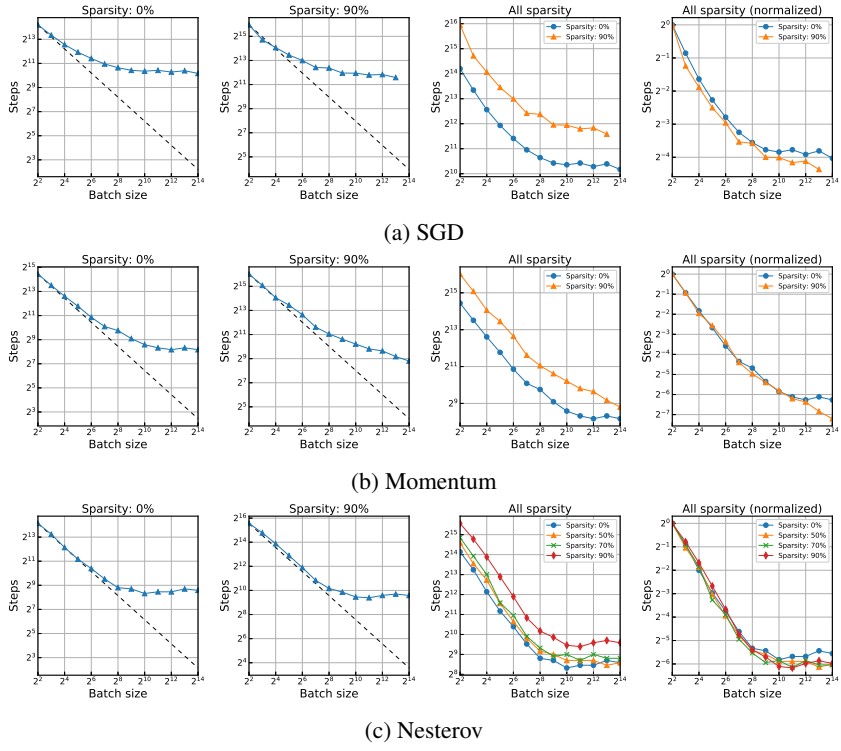

Figure 10: Results for the effects of data parallelism for the workloads of {CIFAR-10, ResNet-8, SGD/Momentum/Nesterov} with a linear learning rate decay.

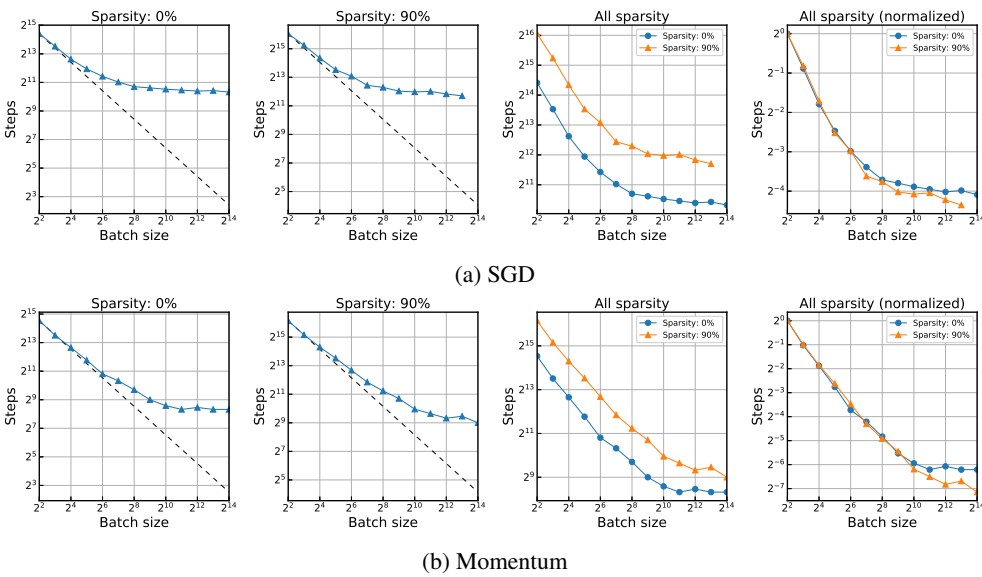

Figure 11: Results for the effects of data parallelism for the workloads of {CIFAR-10, ResNet-8, SGD/Momentum} with a constant learning rate.

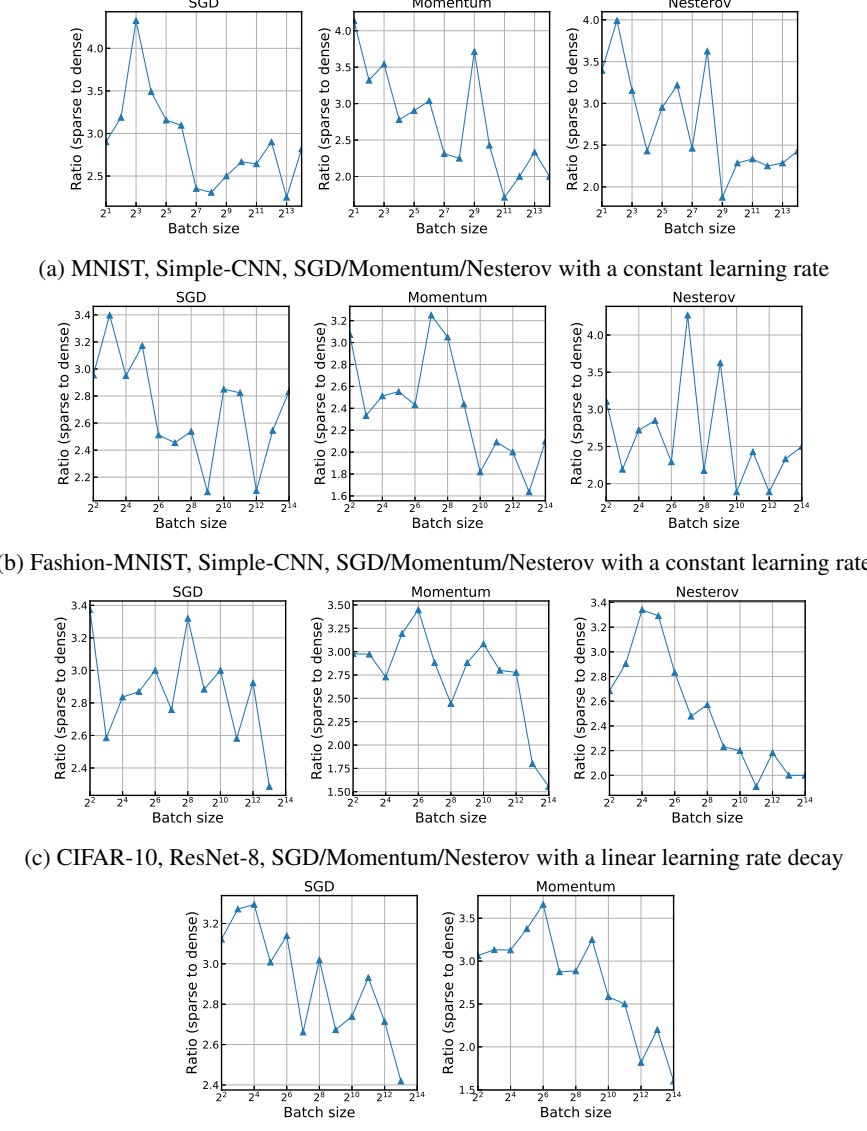

(a) MNIST, Simple-CNN, SGD/Momentum/Nesterov with a constant learning rate

(b) Fashion-MNIST, Simple-CNN, SGD/Momentum/Nesterov with a constant learning rate

(c) CIFAR-10, ResNet-8, SGD/Momentum/Nesterov with a linear learning rate decay

(d) CIFAR-10, ResNet-8, SGD/Momentum with a constant learning rate

Figure 12: Differences in ratio between (90%) sparse network's steps-to-result to dense network's, across different batch sizes for all workloads presented in this work. The difference ranges between (1.5, 4.5) overall. Note that the ratio difference $> 1$ indicates that it requires more number of training iterations (*i.e.*, steps-to-result) for sparse network compared to dense network. Also, the difference seems to decrease as batch size increases, especially for Momentum based optimizers. This potentially indicates that sparse neural networks can benefit from large batch training, despite the general difficulty therein.

## D.2 METAPARAMETER SEARCH RESULTS

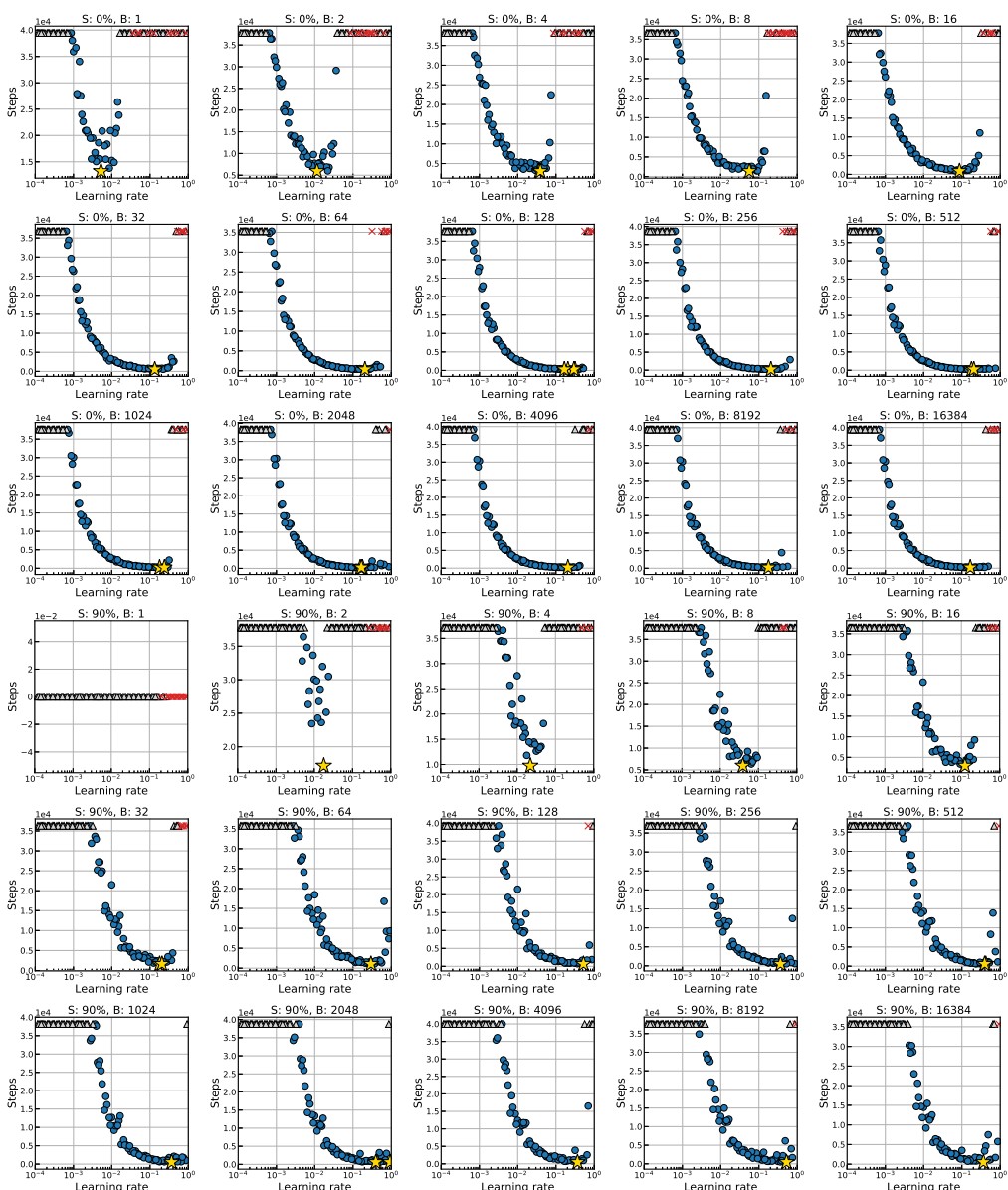

Figure 13: Meataparameter search results for the workloads of {MNIST, Simple-CNN, SGD} with a constant learning rate.

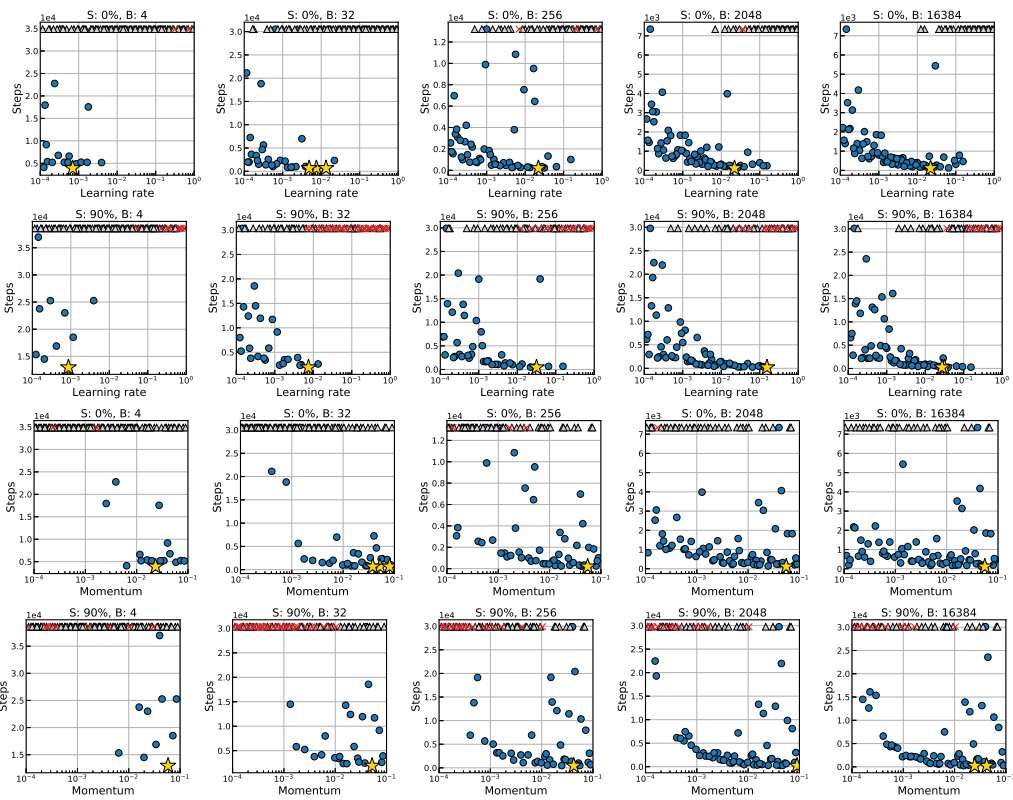

Figure 14: Meataparameter search results for the workloads of {MNIST, Simple-CNN, Momentum} with a constant learning rate.

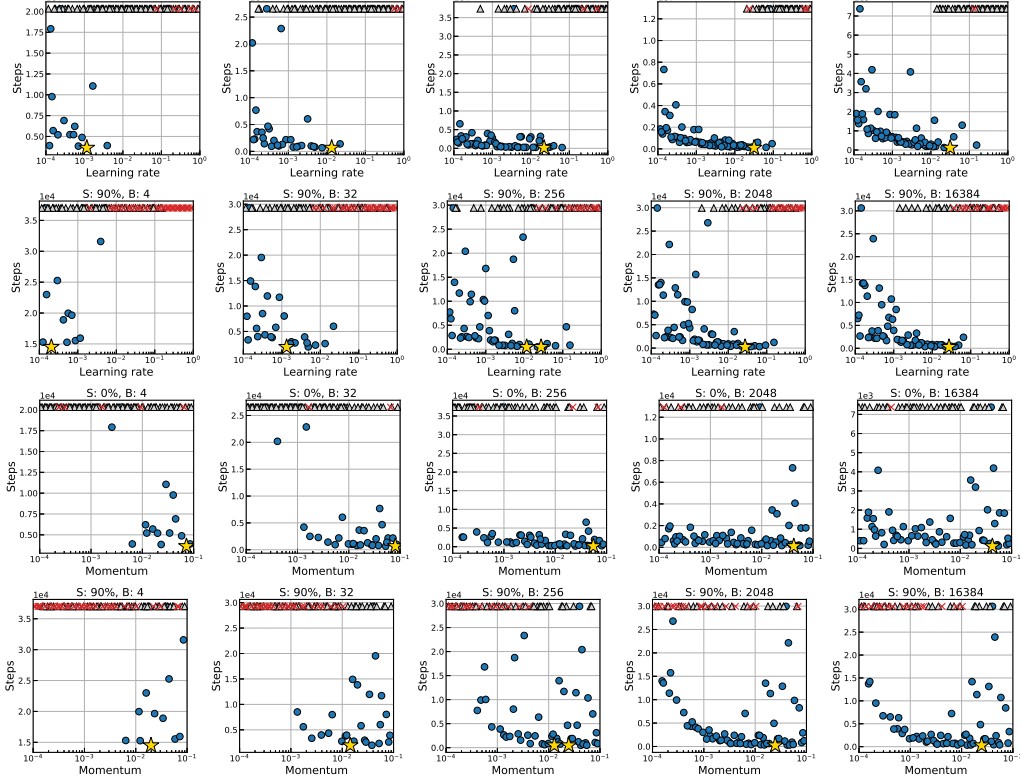

Figure 15: Meataparameter search results for the workloads of {MNIST, Simple-CNN, Nesterov} with a constant learning rate.

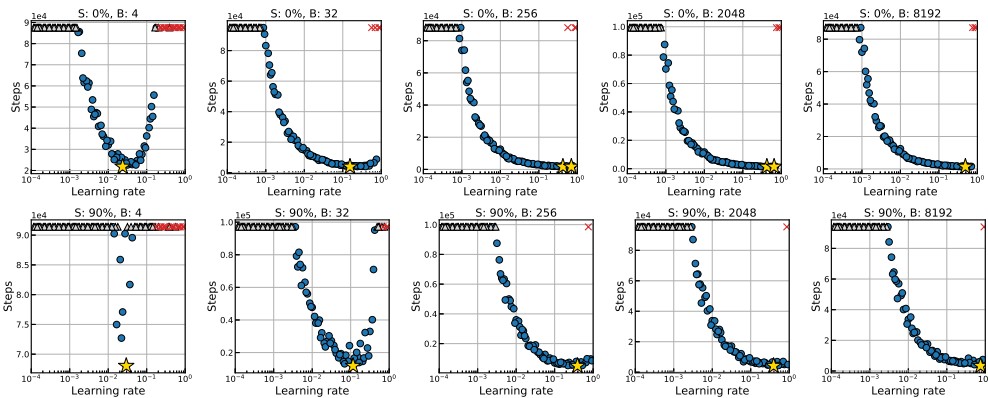

Figure 16: Meataparameter search results for the workloads of {CIFAR-10, ResNet-8, SGD} with a constant learning rate.

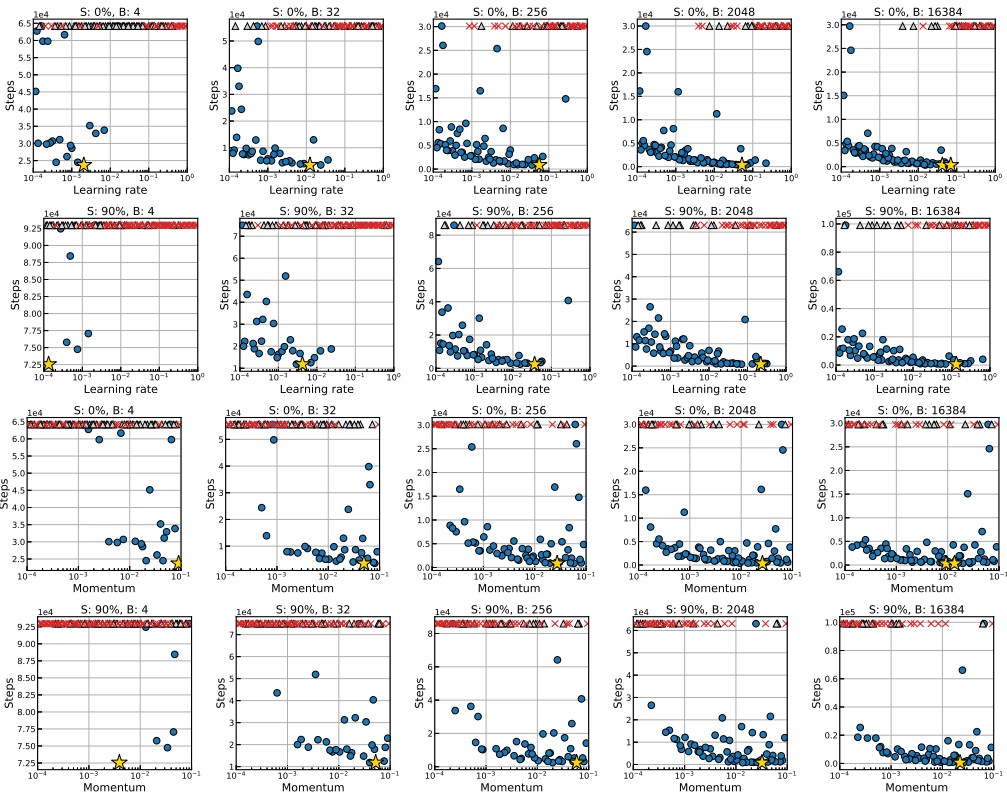

Figure 17: Meataparameter search results for the workloads of {CIFAR-10, ResNet-8, Momentum} with a constant learning rate.

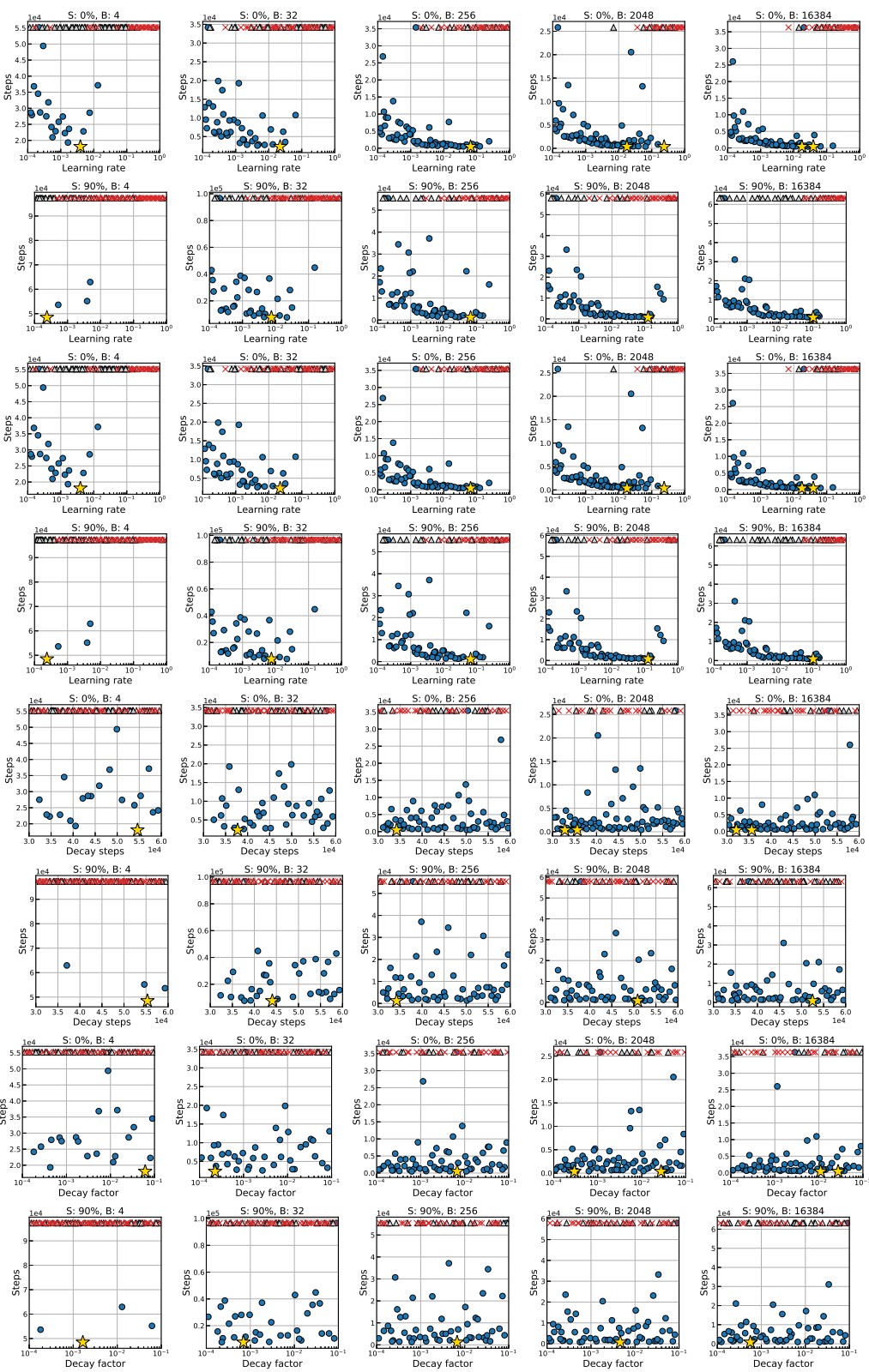

Figure 18: Meataparameter search results for the workloads of {CIFAR-10, ResNet-8, Nesterov} with a linear learning rate decay.

# E IMPLEMENTATION DETAILS

**Data parallelism and sparsity**. By data parallelism, we refer to utilizing a parallel computing system where the training data is distributed to multiple processors for gradient computations, so that the training process can be accelerated. For the purpose of this work, we consider the simplest setting of synchronized distributed systems, in which the degree of parallelism equates to the size of mini-batch used for training on a regular single-node system. This effectively means that the effect of data parallelism can be measured by increasing batch size. By sparsity, we refer to pruning parameters in a neural network model, such that the remaining parameters are distributed sparsely on the network. For the purpose of this work, we employ a recent pruning-at-initialization method to obtain sparse networks, since they must not undergo any training beforehand so as to measure the effects of data parallelism while training from scratch.

**Software and hardware**. We used TensorFlow libraries (Abadi et al., 2016) and a compute cluster with multiple nodes of CPUs (Intel Xeon Gold 5120 CPU @ 2.20GHz with 28 cores; 4 in total) and GPUs (Tesla P100 and V100; 16GB; 28 in total).

