# OpenReview forum: "Understanding the effects of data parallelism and sparsity on neural network training"
_ICLR.cc/2021/Conference — ICLR 2021 Poster_

### Official Review · AnonReviewer2 · 2020-10-27
**Interesting theoretical and empirical study of the interplay of batch size on the required number of optimisation steps at different pruning levels**

**Rating:** 7
**Confidence:** 3

**Review:**

1) Summary
The manuscript studies the effect of batch size at different sparsity levels (achieved by applying connection sensitivity pruning) on the required number of optimisation steps to reach a certain accuracy. The goal is to understand the interplay between those fundamental parameters. The empirical evaluation is performed for different triples of dataset, network architecture and optimisation scheme. The theoretical analysis is based on established bounds on the expected gradient norm.

2) Strengths
+ The paper is well written, the figures and the structure of the text is clear and the experimental setup is concise.
+ The empirical evaluation is very exhaustive and conclusive and allows to draw precise conclusions.
+ Code will be released to reproduce the results. Data sets are public domain.
+ The theoretical analysis is enlightning.

3) Concerns
- While the results are very interesting, the authors could have been more explicit on how the results of this work could help the ordinary neural network user setting parameters in practice by suggesting e.g. a rule of thumb.

4) Remarks/Questions
  a) I found the last sentence of the Introduction a little bold.
  b) "Data parallelism" is a synonyme for "batch size" and "sparsity" is equivalent to "pruning". This could be made more clear in the abstract already.

---

> ### Author Response · Authors · 2020-11-18
> **Response to R2**
>
> We very much appreciate the positive feedback. We address the comments as below, however, if there remain any concerns, please let us know. We will make our best efforts to address them further.
>
> **Rule of thumb**
> While determining an exact critical batch size remains an open question and needs to be accompanied by taking available resources into account, our theoretical analysis indicates that the optimal batch size should be located in the linear scaling region, and it is neither too small nor too large. Also, one could find an estimate of $K^\star$ by calculating $c_1$ for different batch sizes, which could help practitioners to monitor the training progress without having to perform extensive metaparameter search prior to training. When it comes to training sparse neural networks, one could potentially design a new pruning-at-initialization scheme or introduce different optimization tactics as well, in order to address reduced smoothness on sparse networks. Further, one could adjust the batch size to be larger for sparse networks than for regular networks when using momentum or acceleration.
>
> **Last sentence**
> We have updated the last sentence of Introduction as follows: (_previous_) Hence, our results establish a precise and general account of the effects of data parallelism and sparsity on neural network training. $\rightarrow$ (_new_) Being precise and general, our results could help understand the effects of data parallelism and sparsity on neural network training.
>
> **Data parallelism and sparsity**
> We have made these concepts more clear and straightforward as follows: (_Introduction_) For the purpose of this work, we equate data parallelism and sparsity to increasing batch size and pruning model parameters, respectively; we explain these more in detail in Appendix E. (_Appendix E_) By data parallelism, we refer to utilizing a parallel computing system where the training data is distributed to multiple processors for gradient computations, so that the training process can be accelerated. For the purpose of this work, we consider the simplest setting of synchronized distributed systems, in which the degree of parallelism equates to the size of mini-batch used for training on a regular single-node system. This effectively means that the effect of data parallelism can be measured by increasing batch size. By sparsity, we refer to pruning parameters in a neural network model, such that the remaining parameters are distributed sparsely on the network. For the purpose of this work, we employ a recent pruning-at-initialization method to obtain sparse networks, since they must not undergo any training beforehand so as to measure the effects of data parallelism while training from scratch.

---

### Official Review · AnonReviewer3 · 2020-10-28
**The paper presents a practical and theoretical evaluation of the impact of data parallelism and sparsity on neural network training. Extensive measurements confirm the practical / earlier general knowledge, while the theoretical results seems novel.**

**Rating:** 5
**Confidence:** 3

**Review:**

The paper presents both practical (based on measurements) and theoretical result on the impact of data parallelism and sparsity on the training of neural networks.

The practical results are based on extensive measurements of different combination of three datasets, two network models, and four optimization algorithms. Further, the batch size is varied from 2 up to 16384 and the sparsity between 0% and 90%.

In find the practical results interesting and thorough, and they cover a large part of the design space in order to give a good picture of the impact of data parallelism and sparsity. The downside is that the results are in line what is expected when varying data parallelism and sparsity. Exactly where the different plateaus are and the exact slope of the curves varies of course with the selected datasets and network, but the main picture is as expected.

Regarding the theoretical results I have not been able to dive into details of them, mainly due to my own limitations. However, at a high-level view they seem correct and may have some impact, but I leave it to more competent people judge the novelty and contribution of these results.

Other comments/questions:
* I could find any information about the hardware that you have run your experiments on. Should be included in a revised version.
* The scales of the x-axis in Fig 1 is different in different diagrams, which makes a comparison harder.
* The caption of Fig 1 is far too long, move the contents to the main text instead.
* Very good with an extensive Appendix with a lot of measurements.

---

> ### Author Response · Authors · 2020-11-18
> **Response to R3**
>
> We very much appreciate the positive feedback. We address the comments as below, however, if there remain any concerns, please let us know. We will make our best efforts to address them further.
>
> **Expected results**
> While prior works on data parallelism have shown a similar scaling trend [1,2,3], they remain empirical without solid theoretical evidence to confirm the generality of the phenomenon. The difficulty of training sparse networks has also been noticed recently [4], however, the potential cause of the difficulty remains rather unknown as of yet. On the other hand, we establish theoretical results that precisely account for these phenomena. We would like to consider that our results are quite significant, in that these phenomena, which have only been addressed empirically and thus remained as debatable, are now theoretically verified with more accurate descriptions and applied to general non-convex objectives. Further, our findings that the training difficulty of sparse networks is attributed to reduced Lipschitz smoothness, and that sparse networks can have bigger critical batch sizes, have never been previously disclosed, to our best knowledge. For these reasons, we consider the assessment that "the main picture is as expected" is a bit unfair, and is somewhat over-stretched to be used as a criticism for the whole paper.
>
> **Hardware**
> We have added the following in Appendix E: Our experiments are done on a compute cluster with multiple nodes of CPUs (Intel Xeon Gold 5120 CPU @ 2.20GHz with 28 cores; 4 in total) and GPUs (Tesla P100 and V100; 16GB; 28 in total).
>
> **Scale of x-axis**
> We note that they are different workloads and not supposed to be compared with each other. Nevertheless, if it is required by any reason, we could remove/add the batch size of 2 on MNIST/ {Fashion-MNIST, CIFAR}.
>
> **Caption**
> We have made the caption more concise.
>
> **References**
>
> [1] Measuring the effects of data parallelism on neural network training, Shallue et al. 2019
>
> [2] Which algorithmic choices matter at which batch sizes? Insights from a noisy quadratic model, Zhang et al. 2019
>
> [3] An empirical model of large-batch training, McCandlish et al. 2018
>
> [4] The difficulty of training sparse neural networks, Evci et al. 2019

---

### Official Review · AnonReviewer1 · 2020-10-29
**Interesting theoretical results, however, some empirical claims are drawn without enough evidence**

**Rating:** 7
**Confidence:** 4

**Review:**

After reviewing the authors' response:
The authors have agreed to include missing sparsity level-results and have commented that such results are in line with the trends observed in other experiments.  Furthermore, the authors' response addressed all my questions and concerns, for which I'm raising my score.

========================================

The authors explore the role of data parallelism (i.e., the optimal mini-batch size) and sparsity (via parameter pruning) in deep networks through extensive empirical experiments.  This effectively builds on the work of Shallue et al. (2019), by considering sparsity in addition to the previously explored behavior of data parallelism.  Assuming Lipschitz continuity and boundedness on the squared norm of the gradient, the authors then draw theoretical results exposing the role of data parallelism towards convergence for non-convex objectives (note this thus, of course, means local-optima convergence) for SGD with a fixed learning rate (the result for decaying learning rates is included in the appendix).

The main theoretical result is nice, and Lipschitz continuity in DNNs has been extensively studied recently.  In particular, it brings a theoretical understanding to optimizing the batch size, which has generally been understood in empirical/heuristic terms.  However, the paper makes certain leaps in logic and claims with regards to sparsity based on the provided evidence.  Specifically, in Figure 1 sparsity levels are missing from CIFAR-10 and Fashion-MNIST, and Section 3.2 and Appendix D.2 only consider sparsity \in {0.0, 0.9}.  What happens at
sparsity levels between 0 and 90%?  The paper makes many claims about the behavior of sparsity based experiments largely considering only these two sparsity levels, but there is not enough evidence to make general statements of the
 behavior of hyperparameters as a function of sparsity when only
 considering these two extremes.

Also, important experimental details and definitions are omitted from the paper which are necessary to fully understand these results and the soundness of the conclusions drawn from them.  Specifically:

-What is the budget used for the experiments in Figure 3?  The budget
 is actually a hyperparameter of this experiment, and thus it would be
 important to see whether the linear boundary between
 complete/incomplete exists when the budget is either increased or
 decreased, i.e., does this trend generalize for other budgets?

-The definition of "infeasible" in Figure 3 is unclear.  How can a
 trial diverge?  It seems like, given a budget, a trial would simply remain
 incomplete if the budget expired and it had not reached the goal.

-How were the plots in Figure 2 were generated, i.e.,
 were they first normalized then averaged across datasets?  Were
 sparsity results for 50% and 70% included for both Fashion-MNIST and
 CIFAR-10 (one and both, respectively, were missing for these in
 Figure 1)?

-There seem to be implicit assumptions in Section 4.1, specifically, that $f$ is bounded.  For instance, if $f_{\inf}$ goes to either positive or negative infinity, Equation 2 does not say anything helpful.  Furthermore, for
 non-finite $f$, the  expected average squared norm of the gradient
 could grow at a rate faster than linear K, which would invalidate the
 claim in the paper: "the expected average squared norm of... during
 the first K iterations, indicates the degree of convergence; for
 example, it gets smaller as training proceeds with increasing K."  It is important to be clear and precise for these theoretical results.

Other comments:
-The term "maximal data parallelism," which describes the minibatch
size which minimizes the number of steps to convergece, seems
mislabeled.  shouldn't this be minimal data parallelism?  The smallest
minibatch size should be the most parallelizable disregarding
device-to-host communication latency.

-While it is ok to forward reference much of the experimental details
 to the Shallue paper, please at least list the deep learning package
 and hardware used to run the experiments in the paper.

-"This aligns well with the classic result in
learning theory that large batch training allows using bigger learning
rates" <- please include citation for this claim

-Note that (bounds on) Lipschitz constants in DNNs may estimated via:
Fazlyab, Mahyar, et al. "Efficient and accurate estimation of
lipschitz constants for deep neural networks." Advances in Neural
Information Processing Systems. 2019.

---

> ### Author Response · Authors · 2020-11-18
> **Response to R1 (1/2)**
>
> We very much appreciate the positive feedback. We address the comments as below, however, if there remain any concerns, please let us know. We will make our best efforts to address them further.
>
> **Sparsity levels**
> We omitted some sparsity results, only because 0% and 90% contrast the differences introduced by sparsity most clearly, whereas moderate sparsity levels do not seem to add much new information for its cost (e.g., time, space, clutter). Nevertheless, we ran for 50% Fashion-MNIST, and the result will be included in the paper soon (before November 24, 2020). We need more time to finish 50% and 70% CIFAR-10, but we will be able to include the results in the final manuscript. We hope R1 could consider how expensive it is to measure the effect of data parallelism (please see Appendix B). Also, we possess all the other metaparameter search results already for reported sparsity levels to supplement Section 3.2 and Appendix D.2; we have not added them yet to avoid redundancy, but please let us know if we need to include them in the appendix. We stress that a data parallelism curve obtained is a result of extensive measurements (e.g., R2 "The empirical evaluation is very exhaustive and conclusive", R3 "Extensive measurements confirm the practical / earlier general knowledge"), and it is no coincidence that the curve presents such a general phenomenon across various workloads. In fact, this is precisely why we developed a theoretical analysis, which confirms the generality of the behavior.
>
> **Budget**
> The budget is $100$ trials for each and every study setting (i.e., batch size and sparsity). To be more specific, for a given workload (i.e., a combination of data set, network model, and optimization algorithm), we train for the budget of $100$ trials for each batch size and sparsity level, in order to find the best metaparameters and record the lowest steps-to-result (Appendix B). This is in fact indicated in Figure 3, where there are a total $100$ samples of complete, incomplete, and infeasible cases in each figure of a study setting. We expect that increasing/decreasing the number of trials (budget) can make the distribution of the samples more densely/sparsely, but the general appearance will remain the same regardless, forming two distinctive regions of complete and incomplete runs. While we do not study the shape of the boundary separating these regions, we refer to [1] in which it is derived to be linear as $\eta/(1-\gamma)$.
>
> **Infeasible**
> Infeasible trials are those that resulted in divergent training. This can happen when the learning rate and/or momentum is too high (e.g., loss keeps increasing and eventually diverges), and it is observed a lot more when the network is pruned/sparsified, as seen in Figure 3. This training instability for sparse networks can in fact be explained by our finding that pruning/sparsity decreases Lipschitz smoothness, which causes large changes in gradients and thus, is more vulnerable to diverging under backpropagation during training.
>
> **Figure 2**
> This is the result for the workload (MNIST, Simple-CNN, and SGD/Momentum/Nesterov) and study (batch size (2-16384) and sparsity levels (0, 50, 70, 90%)) settings. Hence, there is no normalization or averaging. We have made this clear in the caption.
>
> **Equation 2**
> Firstly, $f$ is an objective function to minimize (e.g., cross-entropy loss in this work), and thus there is no negative infinity. More importantly, Eq. 2 is a standard result of the convergence theorem [2] -- which holds true for any non-convex objective as long as $0 < \bar{\eta} \le \mu/LM_G$ in expectation -- that we chose to take as the basis of our theory of the effect of data parallelism. We note that this result is developed under the standard assumptions on the differentiability of $f$, Lipschitz continuity of $\nabla f$, and bounded variance of $\nabla f$, all of which are generally used to study the convergence properties of stochastic optimization methods.
>
> **Maximal data parallelism**
> Data parallelism refers to the principle of parallelizing data processing using multiple nodes in a distributed system, so as to accelerate the training process. Note that for the simplest setting of a synchronized distributed system (which we focus on for the purpose of this work), the degree of parallelism equates to the size of mini-batch used for training neural networks on a regular single-node system. Therefore, the term maximal data parallelism means that the size of mini-batch is reached to a point, beyond which increasing the batch size will no longer increase the training speed. We further note that this term is borrowed directly from [1].

---

> > ### Author Response · Authors · 2020-11-18
> > **Response to R1 (2/2)**
> >
> > **Deep learning package and hardware**
> > We used TensorFlow libraries [3] and a compute cluster with multiple nodes of CPUs (Intel Xeon Gold 5120 CPU @ 2.20GHz with 28 cores; 4 in total) and GPUs (Tesla P100 and V100; 16GB; 28 in total). We have included this in Appendix E, and further, will release the full code as well as all our measurements.
> >
> > **Reference for "large batch training allows using bigger learning rates"**
> > The noise in gradient reduces as the size of mini-batch increases, which is the fundamental basis for batch and mini-batch gradient methods compared against stochastic gradient methods [4,5]. This effectively means that the variance of gradient decreases as with increasing batch size, and therefore, based on the learning rate requirement $0 < \bar{\eta} \le 1/LM_G$, where $M_G$ is the variance bound (note that using full gradients it reduces to $1/L$ for a steepest descent method), the large batch training will allow bigger learning rates. We further refer to [6], where it states "Theory suggests that when multiplying the batch size by $k$, one should multiply the learning rate by $\sqrt{k}$ to keep the variance in the gradient expectation constant." We have added these references in the paper.
> >
> > **Fazlyab et al. [7]**
> > [7] address Lipschitz constant of $f$, whereas we address Lipschitz constant of $\nabla f$ (i.e., $L$-smoothness). There exist various methods to estimate Lipschitz continuity/smoothness, and yet, these two are measured differently in practice. Our approach is taken from [8] (Appendix C) which is based on [9].
> >
> > **References**
> >
> > [1] Measuring the effects of data parallelism on neural network training, Shallue et al. 2019
> >
> > [2] Optimization methods for large-scale machine learning, Bottou et al. 2018
> >
> > [3] Tensorflow: A system for large-scale machine learning, Abadi et al. 2016
> >
> > [4] A stochastic approximation method, Robbins and Monro 1951
> >
> > [5] Online learning and stochastic approximations, Bottou 1998
> >
> > [6] One weird trick for parallelizing convolutional neural networks, Krizhevsky 2014
> >
> > [7] Efficient and accurate estimation of Lipschitz constants for deep neural networks, Fazlyab et al. 2019
> >
> > [8] Why gradient clipping accelerates training: A theoretical justification for adaptivity, Zhang et al. 2020
> >
> > [9] How does batch normalization help optimization?, Santurkar et al. 2018

---

### Decision · Program_Chairs · 2021-01-07
**Final Decision**

**Decision:**

Accept (Poster)

**Comment:**

On the positive side, this is a quite nice empirical exploration of the interaction between data parallelism and sparsity for training neural networks. The experiments are broad and detailed. On the negative side, the empirical results recapitulate what would be expected and what has already been seen in the literature, as the authors themselves point out ("We note that our observation is consistent with the results of regular network training presented in (Shallue et al., 2019; Zhang et al., 2019)."). And the theory presented, while it does explain the results nicely, is a trivial reformulation of the standard convergence result given in Equation 2. So while this is an interesting paper and the reviewers rated it positively on average, the sparsity exploration is _much_ more novel than the data-parallelism exploration, and there are significant novelty weaknesses that need to be taken into consideration.